# "Everything had stopped, no meeting, no gathering": Social interactions during the COVID-19 pandemic in the Central African Republic, the Democratic Republic of Congo, and Bangladesh

Chiara Altare [1,2]*, Kwanghyun Kim[2,3], IMPACT CAR Team[4¶], IMPACT DRC Team[4¶], IMPACT Bangladesh Team[4¶], Paul B. Spiegel[1,2]

1 Johns Hopkins Bloomberg School of Public Health, Baltimore, United States of America, 2 Johns Hopkins Center for Humanitarian Health, Baltimore, United States of America, 3 Graduate School of Public Health, Kosin University, Busan, Korea, 4 IMPACT Initiatives, Vernier, Switzerland

¶The complete membership of the author groups can be found in the Acknowledgements
* chiara.altare@jhu.edu

## Abstract

Understanding the spread of COVID-19 in humanitarian and fragile settings is challenging for many reasons, including the lack of data on social dynamics and preventive behaviors during an epidemic. We investigate social interactions in three such settings - Democratic Republic of the Congo (DRC), Central African Republic (CAR), Cox's Bazar (CXB), Bangladesh – and how they changed during the first year of the pandemic. This comparative mixed-methods study uses a representative household survey and focus group discussions or key informant interviews in each site. Descriptive weighted analysis of survey responses was conducted; multivariate logistic regression identified factors associated with changes in social interactions. Thematic analysis was conducted on qualitative data. Nearly all participants had social interactions the day before the survey, although the average number of daily interactions was low. Interactions primarily occurred indoors, at home and without masks. We saw a discrepancy between knowledge about and practice of preventive behaviors. Most respondents reported interacting less often (77.3% CXB, 86.7% CAR, 58.8% DRC) and having shorter meetings (80% CXB, 77.8% CAR, and 47.8% DRC). Reluctance towards the COVID-19 vaccine was a risk factor for non-compliant behaviors in CAR (OR increased frequency = 3.51, 95%CI = 1.41–8.75; OR increased duration = 2.47, 95%CI = 1.15–5.29) and DRC (OR increased duration = 3.06, 95%CI = 1.71–5.49), likely pointing to distrust towards institutional policies. Respondents from IDP communities in DRC were less likely to reduce the frequency of interaction, likely because living conditions did not facilitate physical distancing. Increased knowledge in CXB was associated with compliant behavior (for 1pt-increase: OR increased frequency = 0.47, 95%CI = 0.32–0.68; OR increased duration = 0.46, 95%CI = 0.31–0.69).

**Data availability statement:** Deidentified datasets can be made available upon request to the Johns Hopkins Center for Humanitarian Health (email: humanithealth@jhu.edu) after a data sharing agreement is signed to ensure appropriate use.

**Funding:** This work was supported by the Bureau for Humanitarian Assistance, US Agency for International Development (https://www.usaid.gov/) under Grant Number 720FDA20GR00228. The funders had no role in study design, data collection and analysis, decision to publish, or preparation of the manuscript.

**Competing interests:** The authors have declared that no competing interests exist.

**Abbreviations:** CAR, Central African Republic; CI, Confidence Interval; CXB, Cox's Bazar; DRC, Democratic Republic of the Congo; FGD, Focus Group Discussion; HIC, High Income Country; KII, Key Informant interview; LMIC, Low and middle income country; NPI, Non-pharmaceutical interventions; OR, Odds Ratios; aOR, Adjusted Odds Ratios; SD, Standard Deviation.

Understanding social dynamics is fundamental to predict infectious disease spread, particularly in humanitarian settings. More evidence is needed to understand behaviors influencing disease dynamics and drivers of behaviors, including trust in authorities, social, and economic factors. Peace, community engagement, and reduction of misinformation remain critical for epidemic responses in humanitarian settings.

## Introduction

The COVID-19 pandemic affected all countries since its declaration in March 2020, reaching 655 million reported cases and 6.7 million associated deaths by December 31, 2022 [1]. As the SARS-CoV-2 coronavirus spread through airborne or droplet transmission among people in close contact [2], countries introduced non-pharmaceutical interventions (NPIs), limiting movements, contacts and gatherings to reduce the transmission of the disease. By April 2020, most countries enacted relatively stringent policies such as closing their borders, banning mass gatherings, introducing lockdowns, and closing public and private spaces such as schools, markets, restaurants [3].

Low- and middle-income countries (LMICs) and humanitarian and fragile settings were expected to be the least prepared to respond to the pandemic due to multiple existing vulnerabilities including poorer individual nutrition and health status, precarious living conditions, limited access to water and sanitation, as well as limited access to intensive care and oxygen availability [4]. While guidance recommended adapting NPIs to low-resource settings to account for context-specific risks and vulnerabilities [5], most interventions were, at least initially, implemented in a similar manner as in high-income countries (HICs). Social and economic implications were extensive, from loss of livelihoods, interrupted education to the closure of borders, and stigma and discrimination against asylum seekers and refugees [6–8].

Several modeling studies attempted to predict the spread of SARS-CoV-2 to inform preparedness efforts [9,10]. However, the projected scenarios varied greatly compared to what appeared to be occurring in these settings. Humanitarian and fragile settings experienced extensive SARS-COV-2 transmission as shown by sero-prevalence surveys [11–14] and peaks of excess mortality [15–17], but the recorded numbers of deaths and infections were not as well documented as in HICs. There were many hypotheses as to why this may have been the case, including younger age and lower prevalence of non-communicable diseases as a protective factor, early introduction of NPIs, and different roles of temperature and aridity [18,19].

Modeling the spread of COVID-19 in LMICs as well as humanitarian and fragile settings was challenging for many reasons, including the lack of data on social interactions. Consequently, many models relied on estimates of such data from HICs. Before the COVID-19 pandemic, a limited number of studies described social interactions in Kenya [20], South Africa [21], Bangladesh [22], Uganda [23,24], and Zimbabwe [25]. Only a few studies investigated the role of social interactions in the transmission of airborne diseases. One review highlighted difference in characteristics of social dynamics between LMICs and HICs, including how contacts decline

with age in HICs but not in LMICs, how most of the contacts occur at home in LMICs but outside the household in HICs, and the contrasting effect of gender across income strata [26]. One study investigated the prevalence of risk factors relevant for the transmission of respiratory infections among displaced populations in a humanitarian setting in Somaliland [27] and reported age-assortative contacts and high rates of physical contacts, potentially increasing the disease transmission risks.

Since the COVID-19 pandemic, a limited number of studies have tried to estimate how social interactions changed due to NPIs and other factors in LMICs in Kenya [28], South Africa [29], and in multiple African countries [30]. A qualitative study from Sudan [31] highlighted how adherence to mask wearing was low and mainly driven by movement restrictions. Overall, evidence about social interactions in humanitarian and fragile settings remains scarce. This paper investigated social interactions in three humanitarian and fragile settings – the Democratic Republic of the Congo (DRC), Central African Republic (CAR) and Cox's Bazar (CXB), Bangladesh – and how such social interactions changed during the first year of the COVID-19 pandemic.

## Materials and methods

### Study objectives and design

This paper is a comparative mixed-methods study focusing on COVID-19 knowledge and social interactions during the first year of the COVID-19 pandemic (March 1, 2020 - March 31, 2021). It is one of four components of a larger multi-country project investigating COVID-19 epidemiology, health service utilization, COVID-19 knowledge and social interactions, and health care seeking behavior in DRC [32], CAR [33], and Bangladesh [34].

### Study settings

The study was conducted in three sites where two study partners, Action contre la Faim (ACF) and IMPACT, were present: CXB, Bangladesh; Bangui and surrounding areas, CAR; and Mweso Health zone, North Kivu, DRC. Sites were purposively selected as they represent three different humanitarian and fragile settings, both study partners had ongoing operations and overall access (despite delays and limitations highlighted below). The study sites in DRC and CAR are directly affected by insecurity and internal displacement, and are often labelled as humanitarian settings. Cox's Bazar in Bangladesh may be rather described as a "fragile setting" where pre-existing vulnerabilities were exacerbated by the rapid influx of refugees in 2017 (more details on the study sites are provided below). Such differences should be considered when interpreting results across countries.

### Cox's Bazar, Bangladesh

CXB, located in the Chattogram Division of Bangladesh, is one of the nation's poorest and most vulnerable districts due to environmental, social, and economic challenges. The limited resources were further strained since 2017, when over 900,000 forcibly displaced Myanmar nationals fleeing persecution in Myanmar sought refuge in CXB. Bangladesh has reported one of the highest numbers of confirmed COVID-19 infections in South Asia, with 2.04 million cases as of December 1, 2022 [35]. Contributing factors include the country's high population density (1,265 people per km²), poverty (31.5% of the population lives below the poverty line) [36], and limited governmental response capacity. CXB reported the highest number of cases among the study sites, with 6,072 cases by March 2021 [37]. Additionally, CXB had a higher testing rate than DRC (2,867 tests/100,000 population in the first year of the pandemic) and implemented the most stringent COVID-19 response measures compared to CAR and DRC (Fig 1).

### Mweso Health Zone, North Kivu, DRC

The Mweso health zone hosts approximately 450,000 people and is mainly rural, with Mweso as its main town. It is divided into 22 health areas, each with a health center and Mweso Hospital as the one reference hospital. The North

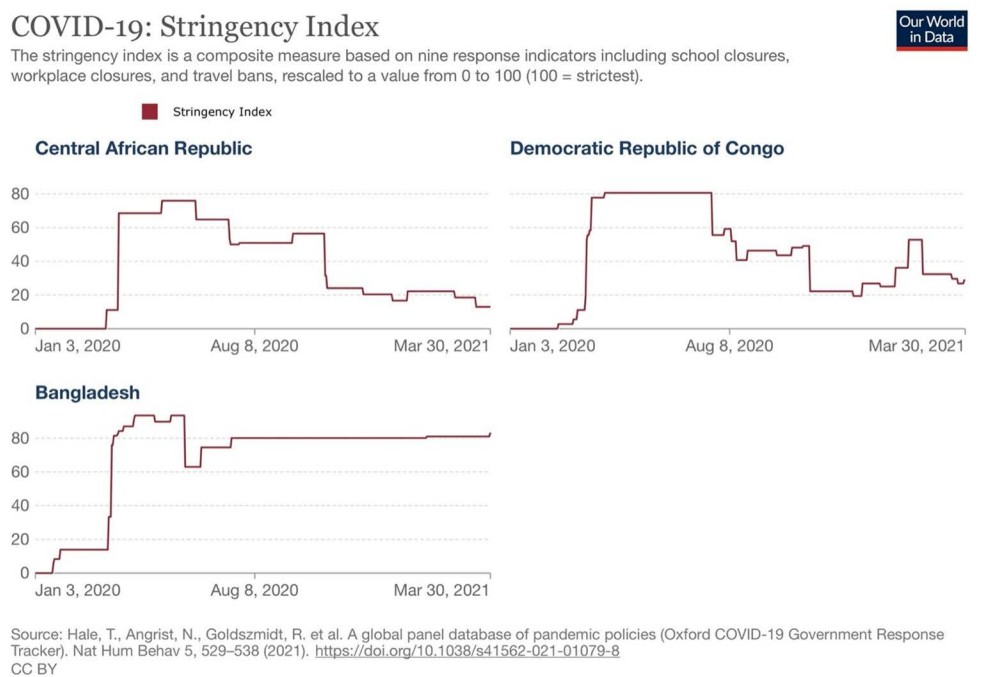

**Fig 1. COVID-19 stringency index by country included in the study [43,44].**

Kivu province has a long history of disease outbreaks, including multiple Ebola, cholera, and measles epidemics [38], and of conflict. Violence and military confrontations have caused extensive suffering, death, and population displacement over many decades. North Kivu province recorded 2,123 confirmed COVID-19 cases and had a testing rate of 73.2 tests/100,000 population as of March 21, 2021 (Table 1). Both testing rate and number of cases were significantly lower than CXB.

## Bangui and surrounding areas, Central African Republic

Since its independence in 1960, CAR has experienced decades of violence and insecurity that have profoundly damaged the economy, infrastructure, and social fabric [39]. While the government retains control of the capital, Bangui, and mostly the western part of the country, its authority is limited in the eastern regions where conflict among armed groups persists, leading to extensive suffering and population displacement. Prior to the onset of COVID-19, life expectancy was among the lowest in the world [40] and mortality was among the highest [41]. Five health districts (Bangui 1, Bangui 2, Bangui 3, Bimbo, and Begoua), mainly urban and peri urban with a population of 1.15 million were included in the study. During the study period, 6,316 COVID-19 cases were reported nationally (Table 1); no separate case count was available for Bangui.

## Study participants, sampling approach, and data collection

Primary qualitative and quantitative data were collected in all sites among adult (>18 years of age) residents and internally displaced persons (IDPs). Three instruments were developed: a questionnaire for the household survey, an interview guide for Key Informant interviews (KII) in Bangladesh, and a guide for the Focus Group Discussions (FGDs) in CAR and DRC. All instruments were developed in English and French and translated as needed into Sango, Swahili, and Bangladeshi. The instruments included questions about social interactions (e.g., occurrence, frequency, location, counterpart, duration) and how they evolved during the COVID-19 pandemic. Another set of questions revolved around knowledge

**Table 1. COVID-19 case counts, incidence rate, and testing rate by study site, March 1, 2020, to March 31, 2021.**

|  | Case Count | Incidence Rate (per 100,000) | Testing Rate (per 100,000) |
|---|---|---|---|
| North Kivu, DRC [32] | 2,213 | 23.15 | 73.2 |
| Cox's Bazar, Bangladesh [34] | 6,072 | 202.65 | 2,867.1 |
| CAR (National) [33] | 6,316 | 68.9 to 81.1[a] | NA[b] |

[a]As the National Laboratory and the Institut Pasteur managed two separate COVID-19 line lists during the first year of the pandemic, two estimates of incidence rates have been calculated [42];

[b]Testing rate at the national level could not be calculated as testing data were unavailable from Institute Pasteur.

of COVID-19 transmission routes, vulnerable groups, symptoms, and willingness to be vaccinated. Detailed notes were taken for the KIIs in CXB and for the FGDs in DRC and CAR by a designated note taker. At the end of interviews or FGD the team debriefed and confirmed notes. Notes were taken directly in English or French by bilingual research assistants.

### Cox's Bazar, Bangladesh

Details about the primary qualitative and quantitative data collection have been previously published [35]. A household survey was conducted by IMPACT (January 13–27, 2022) using a stratified simple random sampling by upazila. The sample size (parameters: 10% margin of error at upazila level, 95% confidence interval, population's proportion with variable of interest of 50%, and 20% non-response/contingency buffer) was 842 households. Individual households were selected via random allocation of a global positioning system (GPS) point within the boundaries of the lowest administrative division (unions). Qualitative data were collected via 23 telephonic KII as in-person meetings were not allowed at the time of data collection (June to October 2021). ACF and IMPACT identified the respondents to include a variety of profiles (e.g., community and religious leaders, elderly, youth, and housewives), age groups (i.e., 18–29, 30–59, 60 + years), and localities (across seven upazilas). Respondents' profiles are available in Table S1 in S1 File.

### Mweso Health Zone, North Kivu, DRC

IMPACT conducted a two-stage cluster sampling household survey in November 4–13, 2021. The sampling frame included 148 settlements larger than five hectares, from which 28 were randomly selected. At least 12 households per settlement were identified using randomly selected GPS points within the boundaries of the settlement. Six settlements were not reachable and were not replaced at random due to logistical and security constraints. Instead, additional interviews were conducted in the nearest accessible settlement. Parameters used to calculate the sample size (657 persons) were +/−5% margin of error, 95% confidence interval, and intra-cluster correlation of 0.06. Twelve semi-structured FGDs were conducted in Swahili between October 27 and November 2, 2021, with 110 participants (community and religious chiefs, elderly, storekeepers, and general community members) who were recruited by IMPACT and ACF teams a few days before the FGD (more details are provided in Table S2 in S1 File).

### Bangui and surrounding areas, Central African Republic

IMPACT conducted a two-stage random sampling household survey in which villages were selected using probability proportional to the size, and households were selected via random allocation of GPS points within the village. The sample size was 1,045 households (calculated with 5% margin of error, 95% confidence interval, population's proportion with variable of interest 50%). Qualitative data originated from 24 semi-structured FGDs, including 192 participants (community members, local and religious leaders, youth and women, and merchants), ensuring a diverse participation. FGDs were stratified by age, sex, and displacement status. Details about the primary qualitative and quantitative data collection have been previously published [42]. More details about the primary data collection are in Table S3 in S1 File.

## Analysis

Notes from the KII/ FGDs were recorded by question. From the notes, unique discussion points raised during the interviews/FGDs were listed under each question and counted to identify frequent topics. A saturation matrix organized by question was used to map findings and confirm the achievement of saturation. Findings were summarized by themes, mostly aligning with the questionnaire structure, first by country and then across study sites. Quotes were extracted and assigned to themes. Data were managed in excel. Additional quotes and themes that emerged from the transcripts outside of the pre-identified questions were recorded.

Knowledge of COVID-19 was assessed in three domains: route of propagation, susceptible population, and preventive measures. Participants were classified into four subgroups according to their level of knowledge about COVID-19: *not informed*, *a little informed*, *informed*, and *well-informed*. Details about the knowledge classification can be found in Table S4 in S1 File.

A descriptive weighted analysis of survey responses was conducted, disaggregated by age, sex, residence, and displacement status of respondents. For continuous variables, weighted t-tests were done for comparison; for categorical variables, weighted χ2 tests were conducted. Multivariable logistic regression was used to investigate factors associated with a change in social interactions: age, sex, displacement status, residence, setting, religion, education, profession, and COVID-19 related knowledge were included as independent variables of the regression model. Since all independent and dependent variables were categorical, there was no need for linearity assumption, normality assumption, and assumption of equal variances. For 'willingness to be vaccinated' and 'knowledge about COVID-19', odds ratios (OR) for changes in frequency and duration of social interaction by each category were calculated, and OR by 1 point increase. P-values less than 0.05 were considered statistically significant. Wald test was conducted to test the global null hypothesis. Analysis was conducted using SAS software version 9.4 [45].

## Ethics approval and consent to participate

JHSPH's IRB determination note 15447 approved primary data collection. Ethical approval from in-country IRB was obtained from the Institute of Health Economics of the University of Dhaka, Bangladesh (IRB letter dated March 28, 2021); the Ethical and Scientific Committee of the University of Bangui on May 17, 2021; the IRB of the University of Bukavu (Letter UCB/CIES/NC/005/ 2021). Participation in the surveys, KII, and FGDs was voluntary, and only adult participants who provided oral informed consent were included. Oral consent was documented in the data collection tool.

Additional information regarding the ethical, cultural, and scientific considerations specific to inclusivity in global research is included in the Supporting Information (S2 Checklist).

## Results

### Quantitative results

Descriptive characteristics of survey participants are presented in Table S5 in S1 File. Most respondents were women in CAR and DRC; participants older than 60 years made up 5.7%, 12%, and 10.2% of the sample in CXB, CAR, and DRC, respectively. Most respondents lived in urban areas in CXB and CAR, while respondents were mainly rural in DRC. Twenty percent of respondents had no education in CXB, 11% in CAR, and 45% in DRC.

Table 2 shows results regarding knowledge, willingness to be vaccinated, and social interaction characteristics. Mean and SD of respondent age in years were 36.70 (SD 11.84) in Bangladesh, 40.01 (SD 13.99) in CAR, and 35.62 (SD 14.40) in DRC. Most of the study participants fell in the category of *little informed* in each country. DRC had the highest proportion of respondents falling in the *not informed* group and the lowest of *well-informed* respondents.

Almost all participants were willing to be vaccinated in CXB (98%), 75% in CAR, and 50% in DRC (combining the two answer options "certainly" and "probably"). One-fifth of the respondents in DRC stated that they would certainly not get the COVID-19 vaccine if it were available.

All study participants in DRC and CAR had social interactions the day before the survey, while 3% of the CXB respondents had none. While the highest average number of interactions per person was in CAR, the duration was similar across countries, with most interactions lasting between 15 min and 1 hour. Most interactions were indoors (in the respondent's house or someone else's house); in the street and at the market were also frequent locations for interactions in CXB and CAR, and less so in DRC. Places of leisure and restaurants/cafés were not common venues for interactions. In most interactions, nobody wore a mask. Both participants wore masks in 12% of the interactions in CXB and 3% and 4%, respectively, in CAR and DRC.

When asked how their social interactions changed during the COVID-19 pandemic, most people reported meeting less often in the three countries (77.3% CXB, 86.7% CAR, 58.8% DRC). Twice as many people in DRC as in CXB did not change the frequency of interactions. 10% of the respondents in CXB reported meeting more often. Most respondents reported having shorter meetings (80%, 77.8%, and 47.8%, respectively, in CXB, CAR and DRC). One-fourth of the respondents in DRC did not change the duration of their interactions. Between 5 and 10% of the respondents in each country reported having longer meetings.

Risk factors for increasing the frequency of meetings (Table 3) were displacement status (i.e., being internally displaced) in DRC (aOR = 2.21, CI = 1.34–3.64) and reluctance to be vaccinated in CAR ("probably not", aOR = 3.51, 95% CI = 1.41–8.75). Increasing knowledge related to COVID-19 was a protective factor in CXB (for 1 pt increase, aOR = 0.47, 95% CI = 0.32–0.68).

Risk factors for increasing the duration of meetings (Table 4) were reluctance to be vaccinated in CAR (aOR = 2.47, 95% CI = 1.15–5.29) and DRC (OR = 3.06, 95% CI = 1.71–5.49). Being female (aOR = 0.57, 95% CI = 0.36–0.91) in CAR and increased knowledge related to COVID-19 in CXB (for 1 pt increase, aOR = 0.46, 95% CI = 0.31–0.69) were protective factors. In all regression models, Wald test indicated that the global null hypothesis could be rejected. (p-value < 0.001)

## Qualitative results

Findings were described around following themes: attitude and knowledge about COVID-19, knowledge and practice of preventative measures, perceptions about vaccines, information sources, and changes in social interactions. Findings by theme and country are available in the case study reports [32–34]. Here, we present only summary findings across countries and quotes complementing the quantitative analysis.

**Attitude and knowledge about COVID.** Most respondents had heard of COVID-19 and believed it was real. Yet, rumors about its existence, origin, prevention and treatment spread in all countries. The government, elites or foreigners were blamed for the introduction of the disease.

*"Community members think that this disease is a trick of the whites to eliminate the blacks because the disease did not appear here in [DRC]"*

*(FGD women, 18–30 rural, DRC)*

**Knowledge and practice of preventive measures.** Mask-wearing, hand washing, and maintaining physical distance were known preventive measures, yet compliance varied across sites. The mentioned reasons included level of enforcement (by police or army), feasibility (maintaining physical distance at the market was considered impossible or breathing with a mask was perceived as difficult), financial constraints, social perception of the severity of the situation, and time.

*"People's perceptions depend on corona's effect on the community people. If our society were affected by corona, then people would be more aware of it. Coronavirus has spread into the city areas of the country. In our village, there is no such situation. So, people are not following the measures. As a result, the present situation is getting worse"*

*(Male retired NGO staff, age 60–69, rural, CXB)*

**Table 2. COVID-19 Knowledge and characteristics of social interactions during the first year of the COVID-19 pandemic in the three study sites.**

| Weighted mean (SD) or N (weighted %) | CXB (N=842) | CAR (N=1,045) | DRC (N=657) | p-value |
|---|---|---|---|---|
| **Sex, N (weighted %)** | | | | <0.001 |
| Men | 443 (52.55) | 409 (37.28) | 219 (33.33) | |
| Women | 398 (47.32) | 636 (62.72) | 438 (66.67) | |
| Other | 1 (0.13) | 0 (0) | 0 (0) | |
| **Age, years, N (weighted %)** | | | | <0.001 |
| 18 – 29 | 249 (29.19) | 271 (26.54) | 263 (40.03) | |
| 30 – 59 | 545 (65.13) | 662 (61.43) | 327 (49.77) | |
| 60 or older | 48 (5.68) | 112 (12.03) | 67 (10.20) | |
| **Knowledge about COVID-19, N (weighted %)** | | | | <0.001 |
| Not informed | 17 (1.98) | 76 (3.53) | 64 (9.74) | |
| A little informed | 479 (57.22) | 601 (53.86) | 392 (59.67) | |
| Informed | 297 (35.04) | 339 (38.46) | 193 (29.38) | |
| Well informed | 26 (3.02) | 29 (4.15) | 4 (0.61) | |
| N/A | 23 (2.74) | 0 (0) | 4 (0.61) | |
| **Willingness to be vaccinated (Bangladesh), N (weighted %)** | | | | |
| Yes-definitely | 807 (98.41) | | | |
| Not yet decided | 13 (1.59) | | | |
| **Willingness to be vaccinated (CAR & DRC), N (weighted %)** | | | | <0.001 |
| Certainly | | 327 (32.20) | 113 (17.23) | |
| Probably | | 539 (43.85) | 218 (33.23) | |
| Neutral | | 52 (6.80) | 108 (16.46) | |
| Probably not | | 57 (6.87) | 76 (11.59) | |
| Certainly not | | 70 (10.28) | 141 (21.49) | |
| **Social interaction, N (weighted %)** | | | | <0.001 |
| Yes | 820 (97.35) | 1,045 (100.00) | 657 (100.00) | |
| **Number of interactions yesterday, weighted mean (SD)\*** | 1.80 (1.18) | 3.50 (1.19) | 2.23 (1.07) | <0.001 |
| **Interaction included physical contact** | 11.28 (1.00) | 91.58 (0.98) | 90.91 (23.55) | <0.001 |
| **Interaction was indoor (vs outdoor), % (SD)** | 41.30 (1.60) | 34.33 (1.47) | 61.59 (2.32) | <0.001 |
| **Location of interaction, % (SD)** | | | | <0.001 |
| My home | 42.60 (1.63) | 42.24 (1.91) | 51.92 (2.54) | |
| Another home | 15.66 (1.16) | 22.83 (1.55) | 10.05 (1.45) | |
| In the street | 17.91 (1.93) | 7.40 (0.84) | 9.98 (1.41) | |
| Shop/market | 15.36 (1.11) | 7.36 (0.94) | 1.64 (0.58) | |
| At work | 4.29 (0.63) | 6.03 (0.98) | 5.65 (1.06) | |
| Place of worship | 0.74 (0.24) | 2.21 (0.38) | 1.00 (0.45) | |
| Place of leisure | 0.87 (0.29) | 3.55 (0.59) | 0.61 (0.35) | |
| School | 0.85 (0.31) | 0.89 (0.38) | 1.00 (0.44) | |
| Community building (e.g., health center) or governmental office | 0.69 (0.23) | 0.92 (0.34) | | |
| Public transport (e.g., bus, shared tom-tom rides) | 0.67 (0.21) | 0.17 (0.10) | | |
| Restaurant/café (including street cafes) | 0.19 (0.10) | 0.47 (0.18) | | |
| Private transport (car, taxi) | 0.12 (0.08) | 0.40 (0.18) | | |

*(Continued)*

**Table 2.** (Continued)

| Weighted mean (SD) or N (weighted %) | CXB (N = 842) | CAR (N = 1,045) | DRC (N = 657) | p-value |
|---|---|---|---|---|
| Other | 0.12 (0.12) | 3.05 (1.42) | 0 (0) | |
| Do not know/ prefer not to respond | | 0.28 (0.14) | 0 (0) | |
| **Duration of interaction, % (SD)** | | | | <0.001 |
| Less than 15 minutes | 59.31 (1.54) | 18.31 (1.51) | 24.49 (2.16) | |
| 15 minutes to an hour | 36.07 (1.47) | 45.61 (1.72) | 45.33 (2.65) | |
| 1–4 hours | 3.83 (0.58) | 28.64 (1.60) | 13.35 (1.65) | |
| more than 4 hours | 0.79 (0.29) | 7.35 (1.04) | 3.47 (0.83) | |
| **Wearing a mask during social interaction, weighted mean (SD)** | | | | <0.001 |
| No - neither of us | 75.32 (1.43) | 95.09 (0.88) | 89.89 (1.38) | |
| Yes - both of us | 12.11 (1.07) | 3.16 (0.77) | 4.13 (0.91) | |
| Yes - only me | 9.69 (0.96) | 0.79 (0.37) | 2.61 (0.72) | |
| Yes - only the contact | 2.88 (0.49) | 0.94 (0.27) | 1.20 (0.49) | |
| **Change in frequency of social interaction, N (weighted %)** | | | | <0.001 |
| Met less often | 633 (77.34) | 863 (86.73) | 386 (58.75) | |
| No change | 103 (12.59) | 142 (8.55) | 157 (23.93) | |
| Met more often | 84 (10.07) | 37 (4.30) | 1 (0.15) | |
| N/A | 0 (0) | 3 (0.42) | 113 (17.20) | |
| **Change in duration of social interaction, N (weighted %)** | | | | <0.001 |
| Met shorter | 657 (80.22) | 748 (77.83) | 314 (47.79) | |
| No change | 120 (14.71) | 177 (11.76) | 166 (25.27) | |
| Met longer | 43 (5.07) | 107 (9.51) | 38 (5.78) | |
| N/A | 0 (0) | 13 (0.90) | 139 (21.16) | |

* Number of interactions in Bangladesh does not include interactions with household members.

"*We favored buying food for children than nose masks*".

*(FGD, Females, urban, 31–59, DRC)*

Opinions regarding compliance with restriction varied across study sites, with some participants sharing how certain interactions and meetings were difficult to avoid (work or shopping); others mentioned that people were continuing to meet for leisure but secretly.

"*People meet in secret despite the restrictions for various reasons relating to work, to buy and sell things in shops, to meet religious leaders, people hide in places where alcohol and beer are sold to drink, people see each other in secret for various meetings.*"

*(FGD, men 31–59, rural, CAR)*

Household size increased across sites due to school closure, movement restrictions, and activity interruptions. Few people speculated this would eventually increase their risk of being infected with COVID-19 in their homes. Furthermore, it was reported that compliance decreased over time as communities became used to COVID-19.

*"These preventive measures changed over time. During the first stage of the lockdown, people did not go out of their houses because of the fear of coronavirus. But recently, it is seen that they don't care about the virus. They are not maintaining social distances either. Earlier, people rarely used to go to shops, but it is normal now. In fact, they go everyday. Because they don't fear corona now. If you look around, you will see similar scenario."*

*(Female teacher, age 20–29, rural, CXB)*

**Perceptions about vaccines.** Regarding vaccines, study participants reported trusting child vaccines. However, they expressed concerns about the COVID-19 vaccine, mainly because of its possible side effects and lack of effectiveness. Several rumors were reported, including that the vaccine can kill people and cause disability or infertility.

*"According to rumors, this vaccine is for white people, they introduced things to hurt us, so we have doubts, it could be that we can also be infected by other viruses"*

*(FGD, men, 18–30, rural, non-displaced, CAR)*

Across countries, participants wished to receive more information about the vaccine and its side effects before deciding whether to be vaccinated.

**Information sources.** Radio, health providers, religious and traditional leaders, national and international NGOs were the most common source of information in DRC and CAR, while radio/TV and social media were mentioned in CXB. Most respondents reported trusting the sources they used, although mistrust on the statistics (number of cases and number of deaths) was reported.

**Changes in social interactions.** Study participants reported a general sense of worry and fear related to meeting people at the beginning of the pandemic, linked to an overall negative perception of the consequences of COVID-19 on social interactions and family bonds.

*"Everything had stopped, no meeting, no gathering, and it had a huge impact on the church, the school, and until today, everyone is paying the price, life had no meaning, we felt like in prison"*

*(FGD men 31–59, rural, CAR)*

Interactions were reduced or suspended entirely in the three sites, where schools and worship venues were closed, and services were interrupted or limited in size. Family members from other villages did not meet for a long time. People's behavior changed:

*"People are meeting with each other less than before. Previously, all the mothers used to sit together and gossip while their children used to play in the field. But this scenario is not common now. Children are playing in their own houses because of COVID-19. People don't visit others if there is no emergency."*

*(Female housewife, age 50–59, CXB, urban)*

Across sites, older people were recognized as being at the highest risk of severe outcomes. Study participants reported being extra careful and reducing interactions with older people to avoid spreading infections. Yet, older study participants lamented feeling discriminated against, left alone, and even despised:

*"The young people no longer want to approach us thinking that we are the most vulnerable"*

*(FGD, mixed, 60+, urban, IDP, CAR)*

*"Certain population groups have contempt for the elderly; they are sidelined"*

*(FGD, Female, 31–59, rural, non-displaced, CAR)*

**Table 3. Factors associated with changes in frequency of social interactions during the first year of the COVID-19 pandemic, by study site.**

| Changes in Frequency: "Meet less often" as reference | Bangladesh | | Central African Republic | | Democratic Republic of Congo | |
|---|---|---|---|---|---|---|
| | aOR (95% CI) | p-value | aOR (95% CI) | p-value | aOR (95% CI) | p-value |
| Sex ('Male' as reference) | 0.76 (0.52 - 1.10) | 0.142 | 0.83 (0.47 - 1.46) | 0.519 | 0.93 (0.55 - 1.54) | 0.765 |
| Age ('18–29' years as reference) | | | | | | |
| 30-59 | 1.33 (0.87 - 2.02) | 0.193 | 1.05 (0.57 - 1.93) | 0.878 | 0.71 (0.45 - 1.11) | 0.029 |
| 60+ | 0.82 (0.32 - 2.14) | 0.688 | 0.93 (0.36 - 2.42) | 0.877 | 1.40 (0.66 - 2.97) | 0.151 |
| Displacement ('Residents' as reference) | – | – | – | – | **2.21 (1.34 - 3.64)** | **0.002** |
| Urban vs. Rural | 0.77 (0.44 - 1.33) | 0.341 | 1.04 (0.52 - 2.06) | 0.911 | 0.53 (0.18 - 1.58) | 0.253 |
| Education ('None' as reference) | | | | | | |
| Primary | 0.85 (0.53 - 1.38) | 0.517 | 1.02 (0.46 - 2.25) | 0.964 | 0.99 (0.59 - 1.67) | 0.974 |
| Secondary | 0.74 (0.44 - 1.26) | 0.271 | 0.80 (0.35 - 1.82) | 0.597 | 1.03 (0.55 - 1.94) | 0.919 |
| Tertiary | 0.54 (0.20 - 1.41) | 0.206 | 2.04 (0.66 - 6.29) | 0.214 | 0.62 (0.33 - 1.43) | 0.626 |
| Willingness to be vaccinated (Bangladesh, 'Definitely' as reference) | 2.74 (0.73 - 10.21) | 0.134 | – | – | – | – |
| Willingness to be vaccinated (CAR & DRC, 'Certainly' as reference) | | | | | | |
| Probably | – | – | 1.28 (0.67 - 2.47) | 0.452 | 0.69 (0.38 - 1.25) | 0.216 |
| Neutral | – | – | 0.53 (0.12 - 2.28) | 0.391 | 0.88 (0.42 - 1.85) | 0.732 |
| Probably not | – | – | **3.51 (1.41 - 8.75)** | **0.007** | 0.42 (0.16 - 1.08) | 0.071 |
| Certainly not | – | – | 1.15 (0.39 - 3.42) | 0.797 | 1.62 (0.87 - 3.02) | 0.130 |
| Knowledge about COVID-19 ('Not informed' as reference) | | | | | | |
| A little informed | 2.02 (0.34 - 11.91) | 0.436 | 1.47 (0.76 - 2.85) | 0.251 | 0.90 (0.46 - 1.74) | 0.753 |
| Informed | 1.06 (0.18 - 6.30) | 0.953 | 1.34 (0.63 - 2.84) | 0.443 | 0.69 (0.33 - 1.43) | 0.320 |
| Well informed | 0.39 (0.04 - 3.71) | 0.413 | * | * | 1.28 (0.10 - 15.98) | 0.848 |
| By 1pt increase | **0.47 (0.32 - 0.68)** | **<0.001** | 0.85 (0.52 - 1.39) | 0.515 | 0.83 (0.58 - 1.20) | 0.324 |

## Discussion

This study analyzes social interactions and their characteristics in three different humanitarian and fragile settings during the COVID-19 pandemic. Several characteristics of the interactions are relevant to disease dynamics. First, the number of contacts a person has during a specific time. While we found a low average number of daily interactions (in absolute terms) across the three sites, it is challenging to interpret our results without baseline or other estimates from the same

**Table 4. Factors associated with changes in duration of social interactions during the first year of the COVID-19 pandemic, by study site.**

| Changes in duration: "Met shorter" as reference | Bangladesh | | Central African Republic | | Democratic Republic of Congo | |
|---|---|---|---|---|---|---|
| | aOR (95% CI) | p-value | aOR (95% CI) | p-value | aOR (95% CI) | p-value |
| Sex ('Male' as reference) | 1.12 (0.76 - 1.65) | 0.561 | **0.57 (0.36 - 0.91)** | **0.018** | 0.81 (0.51 - 1.29) | 0.372 |
| Age ('18–29' as reference) | | | | | | |
| 30–59 | 1.09 (0.70 - 1.69) | 0.703 | 0.71 (0.43 - 1.16) | 0.173 | 1.09 (0.72 - 1.65) | 0.672 |
| 60+ | 1.08 (0.42 - 2.79) | 0.870 | 0.52 (0.23 - 1.17) | 0.112 | 0.69 (0.34 - 1.41) | 0.306 |
| Displacement ('Residents' as reference) | – | – | 0.80 (0.49–1.33) | 0.394 | 1.32 (0.85–2.05) | 0.221 |
| Urban vs. Rural | 0.91 (0.53–1.57) | 0.740 | 1.23 (0.68–2.20) | 0.496 | 1.02 (0.62–1.70) | 0.925 |
| Education ('None' as reference) | | | | | | |
| Primary | 0.85 (0.51 - 1.42) | 0.540 | 0.73 (0.38 - 1.40) | 0.345 | 0.82 (0.52 - 1.31) | 0.413 |
| Secondary | 0.89 (0.51 - 1.55) | 0.673 | 0.84 (0.44 - 1.61) | 0.602 | 0.75 (0.43 - 1.32) | 0.315 |
| Tertiary | 0.76 (0.28 - 2.06) | 0.584 | 0.91 (0.30 - 2.75) | 0.864 | 0.61 (0.09 - 4.33) | 0.941 |
| Willingness to be vaccinated (Bangladesh, 'Definitely' as reference) | 1.20 (0.21 - 6.97) | 0.841 | – | – | – | – |
| Willingness to be vaccinated (CAR & DRC, 'Certainly' as reference) | | | | | | |
| Probably | – | – | 1.41 (0.81 - 2.45) | 0.229 | 1.50 (0.86 - 2.62) | 0.149 |
| Neutral | – | – | 0.91 (0.34 - 2.46) | 0.851 | 1.59 (0.84 - 3.03) | 0.156 |
| Probably not | – | – | 1.22 (0.52 - 2.89) | 0.649 | 0.61 (0.26 - 1.45) | 0.260 |
| Certainly not | – | – | **2.47 (1.15 - 5.29)** | **0.020** | **3.06 (1.71 - 5.49)** | **<0.001** |
| Knowledge about COVID-19 ('Not informed' as reference) | | | | | | |
| A little informed | 1.82 (0.32 - 10.35) | 0.500 | 1.97 (0.83 - 4.71) | 0.126 | 0.77 (0.42 - 1.40) | 0.383 |
| Informed | 0.99 (0.17 - 5.76) | 0.995 | 2.18 (0.88 - 5.39) | 0.093 | 0.56 (0.29 - 1.08) | 0.084 |
| Well informed | 0.34 (0.02 - 4.90) | 0.426 | 0.48 (0.05 - 4.52) | 0.525 | 1.98 (0.15 - 25.59) | 0.600 |
| By 1 pt increase | **0.46 (0.31 - 0.69)** | **<0.001** | 1.12 (0.73 - 1.70) | 0.609 | 0.88 (0.65 - 1.21) | 0.445 |

settings. We identified only one study [30] estimating social interactions in several African countries, including DRC, which we used to interpret our results. This multi-country study relied on two rounds of telephonic surveys conducted during the pandemic and found higher numbers of daily interactions (between 10 and 50 depending on the country and the survey round) than we did. While differences in recording methods and recall biases cannot be excluded, the discrepancy may

also be due to different underlying populations. Our study site in DRC is very rural and remote, while the DRC population in Drobeva et al. [30] is primarily urban and has higher education. More data from rural areas remain needed to better understand social dynamics in remote settings, where social interactions, and consequently the spread of disease may occur differently than in urban areas.

The location of the interaction is a second important factor. Most interactions in our study occurred at home, highlighting how the household is the most likely place where disease transmission might have occurred in these contexts. This finding aligns with existing evidence from LMICs [26] and has implications for the design of response measures. The shutting down of venues where people gather outside of the house, such as shops, restaurants, or other services, may therefore be less effective in reducing transmission in humanitarian and fragile settings compared to HICs where most interactions occur outside of the house [26]. Yet such measures have important consequences on livelihoods. Focusing on individual or household-level measures may yield better results than societal-level interventions in these settings; however, implementing the former has proved challenging. The idea of shielding the most vulnerable (instead of limiting the entire society) was suggested at the beginning of the pandemic [46], with some forms of shielding at either household or extended family levels attempted in Yemen and India [47]. Other experiences focused on feasibility and acceptability assessments (for example, in Sudan, DRC, and Syria [47]) without implementation. The main challenges to the implementation of shielding vulnerable persons in households were its feasibility (e.g., the need for additional support networks outside of households) and acceptability (e.g., segregating older people from their families was questioned [48]). Similar concerns and feelings of loneliness and exclusion were reported by elderly participants in our FGDs, confirming the difficulty of implementing individual or household-level measures. The appropriate mix of measures at the individual and societal level is likely context-specific, and may depend upon several factors, including the existence of a functioning surveillance system, epidemiological, societal, and economic considerations, as well as response capacity [49].

Third, the adoption of preventive measures during interaction, such as wearing a mask and avoiding physical contact, also affects the risk of infection. In our sites, we saw a discrepancy between knowledge about and practice of preventive measures. While most study participants could mention preventive practices, few reported applying them. Respondents from Bangladesh reported a higher use of masks than CAR and DRC, likely due to the more common utilization of masks in Asian countries [50,51]. Mask adherence in African countries has varied and difficult to estimate. Reported levels of compliance ranged from 94% in Mozambique to 43% in DRC and 32% in Uganda [52]. However, most studies were web-based, and thus carried a higher social desirability bias. Barriers to the use of masks vary by context, and include demographic factors, individual risk perception, financial constraints, cultural acceptance and social norms, and personal unease. Identifying such drivers remains instrumental for context-adapted behavior change strategies [53–55].

Regarding behavior changes, the reported frequency and duration of social interactions decreased during the COVID-19 pandemic, aligning with recommended behavior during the pandemic [56]. Our research was not an observational study, and thus we are unable to confirm whether the interactions were shorter and rarer than before COVID-19. The extent to which respondents overreported a behavior is difficult to estimate, but desirability bias cannot be excluded. However, given the low reported use of masks across the three sites and the reporting of non-compliant behaviors, we can presume that responses may reflect practice.

We identified a few factors associated with or hindering recommended behaviors. Reluctance towards the COVID-19 vaccine was a risk factor for non-compliant behaviors in CAR and DRC, likely pointing to distrust towards institutional policies and actions. Dissatisfaction with governmental measures was found to influence non-adherence to public health instructions in other provinces in DRC [57], and it was likely exacerbated in eastern DRC, where tensions with the central authority have characterized the last few decades. Furthermore, the response to the 2018–2020 Ebola outbreak deeply reduced the population's confidence in the authorities [58], limiting the government's capacity to respond to COVID-19, including vaccinating communities. Targeted sensitization and information campaigns are needed to address communities' fears and questions through accessible methods and communication channels.

Unlike with outbreak vaccinations such as Ebola and COVID-19 immunizations, vaccine hesitancy related to routine vaccination for children is generally lower in LMICs than in HICs [59], including DRC [60,61] and possibly CAR (although no data are available). Approaches that leverage these systems and services may thus increase trust for new vaccines. For example, the CAR government increased vaccination coverage in 2022 by tasking existing CHWs (already vaccinating their communities) with the COVID-19 vaccine instead of hiring new CHWs. The government also attempted to understand the causes of hesitancy by engaging with communities to build trust [62]. Rumors about COVID-19 and vaccine hesitancy played a role in shaping behaviors, contributing to overall lower vaccination rates in LMICs than in HICs [63]. Campaigns that are localized and tailored to specific audiences may help to dispel rumors and anxieties. Promising strategies include having local leaders getting vaccinated [64], working with religious organizations to co-design events and messages [65], and encouraging community-level discussions to dispel misunderstanding and doubts [66]. Increasing trust and providing reliable information are likely necessary (although insufficient) factors for changing behaviors. Encouraging are our results from Bangladesh where increased knowledge about COVID-19 was a protective factor.

Finally, respondents from IDP communities in DRC were less likely to reduce the frequency of interaction, likely because their living conditions and site overcrowding did not allow for physical distancing. Similar challenges have been reported in other displacement settings in North Kivu [67] where IDPs advocated for long-term solutions, such as peace, emphasizing how physical distancing could be possible in their place of origin where there is more space than in IDP sites.

While we focus on social interactions during and in relation to the COVID-19 pandemic, it was beyond the scope of the study to explore broader societal dilemmas that arise during particular situations like a pandemic and that affect population dynamics. The need to meet basic needs such as food or to maintain psychological and mental wellbeing may lead to non-compliant behaviors (from a narrow disease prevention perspective) that however may support population resilience in other ways.

Our study has several limitations. We cannot exclude recall bias as data were collected several months after the pandemic's beginning. Yet, restrictions were still in place at the time of data collection, although with lower compliance. Given the impossibility of accessing the study sites at the early stage of the pandemic, we could not use a diary method to track social interactions, which usually provides more reliable results [26]. The number of contacts may, therefore, be underestimated. Furthermore, as repeat surveys were not feasible given access constraints, we could only rely on self-reported changes in behavior during the months of the COVID-19 pandemic with more stringent restrictions. Self-reporting can be subject to desirability bias, and lead to over or under estimation of behavior. Understanding the gap is however difficult given the lack of existing data for comparison. While results have to be interpreted with caution given this limitation, study participants seemed to admit not wearing a mask when interacting with other people, possibly pointing to a low level of desirability bias (i.e., the majority would have reported wearing a mask if they wanted to please the interviewer). We decided to not collect information on symptoms or COVID-19 status due to the non-specificity of symptoms and the lack of rapid diagnostic tests. Therefore, we unfortunately could not estimate the relationship between social interactions and disease spread nor identify risk factors for transmission. Certain villages in Mweso were not accessible due to insecurity at the time of data collection. This may have introduced a bias if populations in the non-accessible areas differ in behavior from the accessible areas. While we cannot completely exclude this, insecurity has affected the entirety of Mweso over time, exposing the entire population to conflict. Finally, study participants were limited to adults, and we did not investigate social interactions among children and youth.

## Conclusions

Understanding social dynamics is fundamental to better predict infectious disease spread. Evidence from humanitarian and fragile settings is, nevertheless, scarce. We studied interactions in three humanitarian and fragile settings and found a low absolute number of interactions, mostly occurring at home. This finding suggests that the most appropriate mix of individual and societal level public health measures needs to be context-specific and consider socio-economic consequences.

Mask wearing was low, while meetings were reported shorter and less frequent. Reluctance to having the COVID-19 vaccine was a risk factor for non-compliant behavior, pointing to low trust in institutions and policies. Risk communication and community engagement is therefore important to dispel rumors and efforts need to be continued and tailored to the specific community. Finally, more evidence is needed to understand behaviors that will influence disease dynamics in future pandemics. Drivers of behavior changes, including trust in authorities, social, cultural, and economic factors, may vary across population groups (residents or IDP) and settings (e.g., rural, urban), and their identification is key to design effective response strategies. Peace, community engagement, and reduction of misinformation remain critical for future epidemic responses in humanitarian and fragile settings.

## Supporting information

**S1 File. Supplementary information: Methods and additional results.**
(DOCX)

**S2 Checklist. Checklist on inclusivity in global research.**
(DOCX)

## Acknowledgments

We would like to thank Marissa Smith, Madison Bates, Natalia Hernandez Morfin, and Sharon Leslie helped analyze the data and draft the case study reports.

Group authorships are as follows:

IMPACT CAR Team
Affiliation: IMPACT CAR Office
Lead author: Melissa Pointet, Samuel Carcanague
Contact email: Katie Rickard, katie.rickard@impat-initiative.org
Group members: Ugo Semat, Melissa Pointet, Sarah McArthur, Samuel Carcanague, Ana Tolosa, Rodrigue Biguioh, Katie Rickard, Roxana Mullafiroze.

IMPACT DRC Team
Affiliation: IMPACT DRC Office
Lead Authors: Jasper Linke, NOortje Gerritsma
Contact Email: Katie Rickard, katie.rickard@impact-initiatives.org
Group members: Roxana Mullafiroze, Jasper Linke, Olivier Cecchi, Nayana Das, Katie Rickard; Jean-Paul Mushamalirwa, Destin Ruhinda, Nadia Lehmann, Marie Amandine, Eliora Henzler, Audrey Gallecier, Benoit Besnardeau, Noortje Gerritsma

IMPACT Bangladesh Team
Affiliation: IMPACT Bangladesh Office
Lead author: Sara Chowdhury
Contact email: Katie Rickard, katie.rickard@impact-initiatives.org
Group members: Roxana Mullafiroze, Jasper Linke, Olivier Cecchi, Nayana Das, Katie Rickard, Rosie Witton, Sara Chowdhury, Zia Foisal, Teresa Schwarz, Glenn Poresh.

## Author contributions

**Conceptualization:** Chiara Altare, Paul B. Spiegel.

**Data curation:** Kwanghyun Kim.

**Formal analysis:** Chiara Altare, Kwanghyun Kim.

**Funding acquisition:** Chiara Altare, Paul B. Spiegel.

**Investigation:** Chiara Altare.

**Methodology:** Chiara Altare, Paul B. Spiegel.

**Project administration:** Chiara Altare, Paul B. Spiegel.

**Supervision:** Chiara Altare, Paul B. Spiegel.

**Validation:** Chiara Altare, Paul B. Spiegel.

**Writing – original draft:** Chiara Altare.

**Writing – review & editing:** Chiara Altare, Kwanghyun Kim, Paul B. Spiegel.

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
