## [Decision Letter · Decision Letter 0]

23 Dec 2024

PONE-D-24-27306“Everything had stopped, no meeting, no gathering”: social interactions during the COVID-19 pandemic in the Central African Republic, the Democratic Republic of Congo, and Bangladesh.PLOS ONE

Dear Dr. Altare,

Thank you for submitting your manuscript to PLOS ONE. After careful consideration, we feel that it has merit but does not fully meet PLOS ONE’s publication criteria as it currently stands. Therefore, we invite you to submit a revised version of the manuscript that addresses the points raised during the review process.

An unmarked version of your revised paper without tracked changes. You should upload this as a separate file labeled 'Manuscript'

We look forward to receiving your revised manuscript.

Kind regards,

Sk Md Mamunur Rahman Malik

Academic Editor

PLOS ONE

**Journal Requirements:**

3. In the ethics statement in the Methods, you have specified that verbal consent was obtained. Please provide additional details regarding how this consent was documented and witnessed, and state whether this was approved by the IRB.

This work was supported by the Bureau for Humanitarian Assistance, US Agency for International Development (https://www.usaid.gov/) under Grant Number 720FDA20GR00228. The funders had no role in study design, data collection and analysis, decision to publish, or preparation of the manuscript.

We would like to thank the United States Bureau for Humanitarian Assistance (USAID/BHA) for funding this work; Marissa Smith, Madison Bates, Natalia Hernandez Morfin, and Sharon Leslie helped analyze the data and draft the case study reports. 

Group authorships are as follows: 

IMPACT CAR Team: Ugo Semat, Melissa Pointet,  Sarah McArthur, Samuel Carcanague, Ana Tolosa, Rodrigue Biguioh

IMPACT DRC Team: Roxana Mullafiroze, Jasper Linke, Olivier Cecchi, Nayana Das, Katie Rickard; Jean-Paul Mushamalirwa, Destin Ruhinda, Nadia Lehmann, Marie Amandine, Eliora Henzler, Audrey Gallecier, Benoit Besnardeau, Noortje Gerritsma

IMPACT Bangladesh Team: Roxana Mullafiroze, Jasper Linke, Olivier Cecchi, Nayana Das, Katie Rickard, Rosie Witton, Sara Chowdhury, Zia Foisal, Teresa Schwarz, Glenn Poresh.

This work was supported by the Bureau for Humanitarian Assistance, US Agency for International Development (https://www.usaid.gov/) under Grant Number 720FDA20GR00228. The funders had no role in study design, data collection and analysis, decision to publish, or preparation of the manuscript.

6. One of the noted authors is a group or consortium "IMPACT CAR Team; IMPACT DRC Team; IMPACT Bangladesh Team". In addition to naming the author group, please list the individual authors and affiliations within this group in the acknowledgments section of your manuscript. Please also indicate clearly a lead author for this group along with a contact email address.

7. We note that there is identifying data in the Supporting Information file "Supplementary material_Soc Interactions_July 3 24". Due to the inclusion of these potentially identifying data, we have removed this file from your file inventory. Prior to sharing human research participant data, authors should consult with an ethics committee to ensure data are shared in accordance with participant consent and all applicable local laws.

-Location data

Reviewers' comments:

Reviewer's Responses to Questions

**Comments to the Author**

1. Is the manuscript technically sound, and do the data support the conclusions?

Reviewer #1: Yes

Reviewer #2: Yes

2. Has the statistical analysis been performed appropriately and rigorously? 

Reviewer #1: No

Reviewer #2: Yes

3. Have the authors made all data underlying the findings in their manuscript fully available?

Reviewer #1: Yes

Reviewer #2: Yes

4. Is the manuscript presented in an intelligible fashion and written in standard English?

Reviewer #1: Yes

Reviewer #2: Yes

5. Review Comments to the Author

**Reviewer #1:**  The paper presents a valuable investigation into how social interactions in three humanitarian settings (Democratic Republic of the Congo, Central African Republic, and Cox’s Bazar, Bangladesh) changed during the first year of the COVID-19 pandemic. This is a significant contribution given the unique challenges and vulnerabilities these settings face.

One of the novel aspects is the comparative mixed-methods approach, combining representative household surveys with qualitative data from focus group discussions and key informant interviews. This methodology provides a comprehensive understanding of social dynamics and preventive behaviors during an epidemic.

The paper highlights the discrepancy between knowledge about and practice of preventive behaviors and identifies factors associated with non-compliant behaviors, such as reluctance towards COVID-19 vaccination and living conditions in IDP communities.

Strengths:

The study addresses a critical gap in understanding social interactions in humanitarian settings during a pandemic, which has been underexplored in existing literature.

The methodology is robust, with a clear explanation of the mixed-methods approach and the rationale for selecting the study sites. The use of both quantitative and qualitative data enriches the analysis.

The findings are relevant and timely, providing insights that could inform future public health interventions in similar settings. For example, the identification of trust in authorities and socio-economic factors as drivers of behavior is crucial for designing effective public health strategies.

The paper is well-organized, with a clear structure that guides the reader through the background, methodology, results, and discussion.

Weaknesses:

The study's reliance on self-reported data could introduce bias, particularly in sensitive topics such as compliance with preventive measures and vaccination attitudes. The paper does not adequately address how this potential bias was mitigated.

The paper could benefit from a more in-depth discussion of the limitations of the study. For example, the authors should discuss the potential impact of the non-reachable settlements in DRC on the representativeness of the data.

Some of the statistical analyses, such as the multivariate logistic regression, are presented without sufficient detail on the assumptions and checks performed to validate the models.

Suggestions for Improvement:

Address the potential biases introduced by self-reported data by discussing any steps taken to validate the responses or by including a section on the limitations of self-reported data. For example, cross-referencing self-reported behaviors with observed behaviors or using triangulation methods could strengthen the validity of the findings.

Expand the discussion on the limitations of the study, particularly regarding the non-reachable settlements in DRC and any other logistical challenges encountered. This transparency will help readers critically assess the robustness of the study.

Provide more detail on the statistical analysis methods, including the assumptions of the logistic regression models and how these were checked. This could include information on multicollinearity tests, goodness-of-fit measures, and any sensitivity analyses performed.

Enhance the discussion on the implications of the findings for public health policy and practice. For example, the paper could suggest specific interventions to increase trust in authorities or address the socio-economic factors influencing preventive behaviors.

**Reviewer #2: ** The study is very important and touches a very critical aspect of population health during pandemic. Please consider the following:

1- Adjust the article and use the past tense

2- Add to the keywords list: pandemic; population health

3- Shed light on the impact of pandemic and borders closure on health system and supply chains

4- Identify the impact of humanitarian aids supported underserved and vulnerable population

5- What are the common characteristics and differences between the three countries? and define the rationale of selecting those countries.

6- Shed light on the interaction between pandemic related psychological dilemma and population dynamic

7- Qualitative analysis is inadequate.

8- In contrary to other population, explain the reasons for higher rates of covid-19 vaccination acceptance.

9- Add a section for limitation and recommendations

6. PLOS authors have the option to publish the peer review history of their article (what does this mean? ). If published, this will include your full peer review and any attached files.

**Do you want your identity to be public for this peer review?** For information about this choice, including consent withdrawal, please see our Privacy Policy .

Reviewer #1: **Yes: ** Olivier Mukuku

Reviewer #2: **Yes: ** WEAM BANJAR

---

## [Author Response · Author response to Decision Letter 1]

7 Feb 2025

Journal Requirements:

and

Authors’ reply: Thank you. We revised the guidelines and updated the paper accordingly

Authors’ reply: Thank you. We added the questionnaire as Supporting Information.

3. In the ethics statement in the Methods, you have specified that verbal consent was obtained. Please provide additional details regarding how this consent was documented and witnessed, and state whether this was approved by the IRB.

Authors’ reply: Yes, verbal consent was approved by JHSPH IRB as well as in country IRBs. Consent was documented on the data collection tool (i.e. interviewer could not proceed to the interview without confirming consent was given).

This work was supported by the Bureau for Humanitarian Assistance, US Agency for International Development (https://www.usaid.gov/) under Grant Number 720FDA20GR00228. The funders had no role in study design, data collection and analysis, decision to publish, or preparation of the manuscript.

Authors’ reply: Please find here the updated funding statement. Thanks for editing it in the online submission form.

This work was supported by the Bureau for Humanitarian Assistance, US Agency for International Development (https://www.usaid.gov/) under Grant Number 720FDA20GR00228. The funders had no role in study design, data collection and analysis, decision to publish, or preparation of the manuscript. There was no additional external funding received for this study.

We would like to thank the United States Bureau for Humanitarian Assistance (USAID/BHA) for funding this work; Marissa Smith, Madison Bates, Natalia Hernandez Morfin, and Sharon Leslie helped analyze the data and draft the case study reports.

Group authorships are as follows:

IMPACT CAR Team: Ugo Semat, Melissa Pointet, Sarah McArthur, Samuel Carcanague, Ana Tolosa, Rodrigue Biguioh; IMPACT DRC Team: Roxana Mullafiroze, Jasper Linke, Olivier Cecchi, Nayana Das, Katie Rickard; Jean-Paul Mushamalirwa, Destin Ruhinda, Nadia Lehmann, Marie Amandine, Eliora Henzler, Audrey Gallecier, Benoit Besnardeau, Noortje Gerritsma; IMPACT Bangladesh Team: Roxana Mullafiroze, Jasper Linke, Olivier Cecchi, Nayana Das, Katie Rickard, Rosie Witton, Sara Chowdhury, Zia Foisal, Teresa Schwarz, Glenn Poresh.

This work was supported by the Bureau for Humanitarian Assistance, US Agency for International Development (https://www.usaid.gov/) under Grant Number 720FDA20GR00228. The funders had no role in study design, data collection and analysis, decision to publish, or preparation of the manuscript.

Authors’ reply: Noted, please find here the updated funding statement (same as in response to point 4). Thanks for editing it in the online submission form.

This work was supported by the Bureau for Humanitarian Assistance, US Agency for International Development (https://www.usaid.gov/) under Grant Number 720FDA20GR00228. The funders had no role in study design, data collection and analysis, decision to publish, or preparation of the manuscript. There was no additional external funding received for this study.

6. One of the noted authors is a group or consortium "IMPACT CAR Team; IMPACT DRC Team; IMPACT Bangladesh Team". In addition to naming the author group, please list the individual authors and affiliations within this group in the acknowledgments section of your manuscript. Please also indicate clearly a lead author for this group along with a contact email address.

Authors’ reply: Please find here the revised Acknowledgements (also added in the revised version of the manuscript):

*****

We would like to thank Marissa Smith, Madison Bates, Natalia Hernandez Morfin, and Sharon Leslie who helped analyze the data and draft the case study reports.

Group authorships are as follows:

IMPACT CAR Team

Affiliation: IMPACT CAR office

Lead author: Melissa Pointet, Samuel Carcanague

Contact email: Katie Rickard katie.rickard@impact-initiatives.org

Group members: Ugo Semat, Melissa Pointet, Sarah McArthur, Samuel Carcanague, Ana Tolosa, Rodrigue Biguioh. Katie Rickard, Roxana Mullafiroze

IMPACT DRC Team

Affiliation: IMPACT DRC office

Lead author: Jasper Linke, Noortje Gerritsma

Contact email: Katie Rickard katie.rickard@impact-initiatives.org

Group members: Roxana Mullafiroze, Jasper Linke, Olivier Cecchi, Nayana Das, Katie Rickard; Jean-Paul Mushamalirwa, Destin Ruhinda, Nadia Lehmann, Marie Amandine, Eliora Henzler, Audrey Gallecier, Benoit Besnardeau, Noortje Gerritsma.

IMPACT Bangladesh Team

Affiliation: IMPACT Bangladesh office

Lead author: Sara Chowdhury

Contact email Katie Rickard katie.rickard@impact-initiatives.org

Group members: Roxana Mullafiroze, Jasper Linke, Olivier Cecchi, Nayana Das, Katie Rickard, Rosie Witton, Sara Chowdhury, Zia Foisal, Teresa Schwarz, Glenn Poresh.

*****

7. We note that there is identifying data in the Supporting Information file "Supplementary material_Soc Interactions_July 3 24". Due to the inclusion of these potentially identifying data, we have removed this file from your file inventory. Prior to sharing human research participant data, authors should consult with an ethics committee to ensure data are shared in accordance with participant consent and all applicable local laws.

• Name, initials, physical address

• Ages more specific than whole numbers

• Internet protocol (IP) address

• Specific dates (birth dates, death dates, examination dates, etc.)

• Contact information such as phone number or email address

• Location data

• ID numbers that seem specific (long numbers, include initials, titled “Hospital ID”) rather than random (small numbers in numerical order)

Authors’ reply: We have submitted a revised version of supplementary material.

Authors’ reply: Noted. We added captions for the supporting information at the end of the manuscript.

Reviewer's Responses to Questions

Comments to the Author

1. Is the manuscript technically sound, and do the data support the conclusions?

Reviewer #1: Yes

Reviewer #2: Yes

2. Has the statistical analysis been performed appropriately and rigorously?

Reviewer #1: No

Reviewer #2: Yes

3. Have the authors made all data underlying the findings in their manuscript fully available?

Reviewer #1: Yes

Reviewer #2: Yes

4. Is the manuscript presented in an intelligible fashion and written in standard English?

Reviewer #1: Yes

Reviewer #2: Yes

5. Review Comments to the Author

Reviewer #1: The paper presents a valuable investigation into how social interactions in three humanitarian settings (Democratic Republic of the Congo, Central African Republic, and Cox’s Bazar, Bangladesh) changed during the first year of the COVID-19 pandemic. This is a significant contribution given the unique challenges and vulnerabilities these settings face.

One of the novel aspects is the comparative mixed-methods approach, combining representative household surveys with qualitative data from focus group discussions and key informant interviews. This methodology provides a comprehensive understanding of social dynamics and preventive behaviors during an epidemic.

The paper highlights the discrepancy between knowledge about and practice of preventive behaviors and identifies factors associated with non-compliant behaviors, such as reluctance towards COVID-19 vaccination and living conditions in IDP communities.

Authors’ reply: Thank you.

Strengths:

The study addresses a critical gap in understanding social interactions in humanitarian settings during a pandemic, which has been underexplored in existing literature.

The methodology is robust, with a clear explanation of the mixed-methods approach and the rationale for selecting the study sites. The use of both quantitative and qualitative data enriches the analysis.

The findings are relevant and timely, providing insights that could inform future public health interventions in similar settings. For example, the identification of trust in authorities and socio-economic factors as drivers of behavior is crucial for designing effective public health strategies. The paper is well-organized, with a clear structure that guides the reader through the background, methodology, results, and discussion.

Authors’ reply: Thank you.

Weaknesses:

The study's reliance on self-reported data could introduce bias, particularly in sensitive topics such as compliance with preventive measures and vaccination attitudes. The paper does not adequately address how this potential bias was mitigated. The paper could benefit from a more in-depth discussion of the limitations of the study. For example, the authors should discuss the potential impact of the non-reachable settlements in DRC on the representativeness of the data.

Authors’ reply: Thank you for raising this point. We have expanded the limitations of the paper highlighting possible biases affecting the study (lines 456 to 465).

Some of the statistical analyses, such as the multivariate logistic regression, are presented without sufficient detail on the assumptions and checks performed to validate the models.

Authors’ reply: Thank you for your comment. We clarified in the method section (lines 227 to 239) that all variables used in the analysis are categorical as we estimated a logistic regression. We also clarified that we did not need to test for linearity, normality and equal variance assumptions as all independent and dependent variables were categorical.

Following your comment, we added a sentence about the use of the Wald test to the logistic regression models to test the global null hypothesis and assess the model validity (line 242). Results from all regression models showed that global null hypothesis could be rejected, (p-value<0.001) indicating that the independent variables included in these models improve the predictability of the results (lines 298-299).

Suggestions for Improvement:

Address the potential biases introduced by self-reported data by discussing any steps taken to validate the responses or by including a section on the limitations of self-reported data. For example, cross-referencing self-reported behaviors with observed behaviors or using triangulation methods could strengthen the validity of the findings. Expand the discussion on the limitations of the study, particularly regarding the non-reachable settlements in DRC and any other logistical challenges encountered. This transparency will help readers critically assess the robustness of the study.

Authors’ reply: Thank you for this suggestion. We expanded the limitation section in the discussion (lines 452 to 473).

Provide more detail

---

## [Editor Report · Decision Letter 1]

21 Feb 2025

PONE-D-24-27306R1“Everything had stopped, no meeting, no gathering”: social interactions during the COVID-19 pandemic in the Central African Republic, the Democratic Republic of Congo, and Bangladesh.PLOS ONE

Dear Dr. Altare,

Thank you for submitting your manuscript to PLOS ONE. After careful consideration, we feel that it has merit but does not fully meet PLOS ONE’s publication criteria as it currently stands. Therefore, we invite you to submit a revised version of the manuscript that addresses the following minor points raised during the final review process.

1. This article, according to the authors, attempted to understand the spread of COVID-19 in humanitarian settings. However, it was not clear on what considerations the Cox's Bazar area of Bangladesh was selected as a site or an area with a humanitarian context! . The Cox's Bazar area of Bangladesh does not meet the criteria to be called an area with a humanitarian crisis just because it is hosting Rohingya refugees. According to the authors, the site was selected for this study as it is the location of the IMPACT project and possibly the site of a larger multi-country project. It is important, therefore, to provide adequate justifications under the methodology section on the selection of Cox's Bazar as a site that is comparable to CAR and the DRC in terms of demographic, socio-economic, and humanitarian characteristics.

2. In the analysis section, it has been mentioned that "thematic analysis was applied to summarize relevant themes across study participants as well as to identify differences across study sites.". In the result section of the qualitative data, no such analysis by thematic areas and comparison by different sites were noted. It is important that the analyzed qualitative data are presented by the main thematic areas of the study, and a summary of the main findings is presented in a way that shows comparison or commonalities between the sites.

3. More description should be provided on how the qualitative data were analyzed. Such as how the data were transcribed, which software was used to record, and code and for transcription of the data. How the consistency and accuracy were checked as the interviews and FGDs were done in three sites in the local language. 

4. It is also important to highlight in the section on strength and limitation if the translation of the KII and FGD into English from the local language introduced any inconsistencies and how they were checked and prevented.

5. For sample size selection, a "prevalence of 50%" was used to select the household number for Cox's Bazar and CAR. It is not clear what this "50% prevalence" refers to. What is the source of such an assumption? The pandemic was a new event. Therefore, what was the reference underlying the author's decision to use a "50% prevalence" for what variable in order to select the right sample? Please provide some clarity in the text to make the readers understand the rationale behind such an assumption.

6. In Table 2, please provide the mean age (SD) or median (IR) age of the respondents by three study sites. It is also important to present the  gender characteristics of the respondents though these are availabel in the supplementary table. 

7. In Tables 3 and 4, we understand that the data are analyzed using a multivariate logistic regression analysis. As such, the Odds Ratios presented in these tables should be named as adjusted Odds Ratios (aOR) unless the authors have other justifications for not naming these ratios as "aOR".

8. It is not clear why (in Table 3) the columns for "Displacement" for Bangladesh and CAR are empty. No displaced people from either of these two countries were included in the sample.? Please provide a justification in the footnote of the table.

9. The author's findings that "IDPs were at higher risk and increased and faced increased barriers to protecting themselves from COVID-19" is not substantiated by data presented in Table 4 (P value not significant). Any clarity will be helpful. 

We look forward to receiving your revised manuscript.

Kind regards,

Sk Md Mamunur Rahman Malik

Academic Editor

PLOS ONE
---

## [Author Response · Author response to Decision Letter 2]

28 Mar 2025

Baltimore March 28, 2025

Response to the journal and reviewers’ comments on the article:

“Everything had stopped, no meeting, no gathering”: social interactions during the COVID-19 pandemic in the Central African Republic, the Democratic Republic of Congo, and Bangladesh.

Dear Editors of Plos One,

Please find below our point-by-point response to the editorial and reviewers’ comments.

We have indicated whether and how we have revised the text and referenced line numbers in the track change version of the revised paper with author details.

We remain at your disposal for any clarification, and we look forward to hearing from you.

With thanks,

Chiara Altare

_______

Reviewer’s comments

1. This article, according to the authors, attempted to understand the spread of COVID-19 in humanitarian settings. However, it was not clear on what considerations the Cox's Bazar area of Bangladesh was selected as a site or an area with a humanitarian context! . The Cox's Bazar area of Bangladesh does not meet the criteria to be called an area with a humanitarian crisis just because it is hosting Rohingya refugees. According to the authors, the site was selected for this study as it is the location of the IMPACT project and possibly the site of a larger multi-country project. It is important, therefore, to provide adequate justifications under the methodology section on the selection of Cox's Bazar as a site that is comparable to CAR and the DRC in terms of demographic, socio-economic, and humanitarian characteristics.

Authors’ reply: Thank you for your comment. We agree that Cox’s Bazar is not a humanitarian emergency simply because it is hosting Rohingya refugees. However, we believe it can be considered a fragile setting given pre-existing vulnerabilities that were exacerbated by the rapid influx of refugees. We attempted to make this differentiation by using “humanitarian and fragile settings” throughout the document. We did notice a few spots where we only referenced it as a humanitarian setting, and we corrected this.

We also edited the methodology as requested clarifying the differences across study sites [lines 116 – 120].

2. In the analysis section, it has been mentioned that "thematic analysis was applied to summarize relevant themes across study participants as well as to identify differences across study sites."

In the result section of the qualitative data, no such analysis by thematic areas and comparison by different sites were noted. It is important that the analyzed qualitative data are presented by the main thematic areas of the study, and a summary of the main findings is presented in a way that shows comparison or commonalities between the sites.

Authors’ reply: Thank you for the comment. We revised the qualitative results section (page 18 to 21) to highlight the following:

- The themes that were identified in the analysis are: Attitude and knowledge about COVID-19, knowledge and practice of preventative measures, perceptions about vaccines, information sources, and changes in social interactions.

- We made reference to the longer version of the results by theme available in the three country reports.

- We reorganized the findings in the paper around the above mentioned themes but in a more succinct way.

3. More description should be provided on how the qualitative data were analyzed. Such as how the data were transcribed, which software was used to record, and code and for transcription of the data. How the consistency and accuracy were checked as the interviews and FGDs were done in three sites in the local language.

Authors’ reply: Thank you for the comment. We provided more details how qualitative data were managed and analyzed on page 10 [Lines 219-225].

4. It is also important to highlight in the section on strength and limitation if the translation of the KII and FGD into English from the local language introduced any inconsistencies and how they were checked and prevented.

Authors’ reply: Thank you. We clarified that KII and FGD notes were directly taken in English / French by the interviewer and note takers, as deemed easier [lines 171-173]. Therefore, we did not have to translate them again, as the core research team could work in both languages.

5. For sample size selection, a "prevalence of 50%" was used to select the household number for Cox's Bazar and CAR. It is not clear what this "50% prevalence" refers to. What is the source of such an assumption? The pandemic was a new event. Therefore, what was the reference underlying the author's decision to use a "50% prevalence" for what variable in order to select the right sample? Please provide some clarity in the text to make the readers understand the rationale behind such an assumption.

Authors’ reply: To calculate the sample size required for a survey, you usually need to define three key parameters: “prevalence of the variable of interest” (i.e. the population’s proportion who would have a certain characteristics or knowledge or behavior), margin of error and confidence level. As in our case we had multiple variables of interest for which no previous estimate was known, we took the conservative approach of 50% that would generate the biggest sample size. See: https://www.calculator.net/sample-size-calculator.html

We changed “prevalence” to population’s proportion with variable of interest in the method section.

6. In Table 2, please provide the mean age (SD) or median (IR) age of the respondents by three study sites. It is also important to present the gender characteristics of the respondents though these are availabel in the supplementary table.

Authors’ reply: We have added information about age and gender characteristics to Table 2. We have also provided the mean and standard deviation of age by region in Results section (lines 271 – 272): “Mean and SD of respondent age in years were 36.70 (SD 11.84) in Bangladesh, 40.01 (SD 13.99) in CAR, and 35.62 (SD 14.40) in DRC.”

7. In Tables 3 and 4, we understand that the data are analyzed using a multivariate logistic regression analysis. As such, the Odds Ratios presented in these tables should be named as adjusted Odds Ratios (aOR) unless the authors have other justifications for not naming these ratios as "aOR".

Authors’ reply: Thank you, we have changed OR to aOR in tables 3 and 4 and in the narrative results page 15.

8. It is not clear why (in Table 3) the columns for "Displacement" for Bangladesh and CAR are empty. No displaced people from either of these two countries were included in the sample.? Please provide a justification in the footnote of the table.

Authors’ reply: In Bangladesh, no information about displacement was investigated during the survey as all respondents were from the host/ resident communities. In CAR, all individuals who were displaced (N = 190) were residing in urban area. Therefore, the displacement category in CAR coincided with the urban category. We were therefore unable to provide aOR of displacement in multivariate regression model.

9. The author's findings that "IDPs were at higher risk and increased and faced increased barriers to protecting themselves from COVID-19" is not substantiated by data presented in Table 4 (P value not significant). Any clarity will be helpful.

Authors’ reply: Thank you for raising this point. Results in table 3 do support the statement that displacement was a risk factor for increasing the frequency of meetings, however you are right that displacement is not statically significant in table 4 (i.e. we could not identify it as a risk factor for increased duration of meetings). We edited the conclusion accordingly [lines 736-738].

---

## [Editor Report · Decision Letter 2]

2 Apr 2025

“Everything had stopped, no meeting, no gathering”: social interactions during the COVID-19 pandemic in the Central African Republic, the Democratic Republic of Congo, and Bangladesh.

PONE-D-24-27306R2

Dear Dr. Altare,

We’re pleased to inform you that your manuscript has been judged scientifically suitable for publication and will be formally accepted for publication once it meets all outstanding technical requirements.

Kind regards,

Sk Md Mamunur Rahman Malik

Academic Editor

PLOS ONE
---

## [Editor Report · Acceptance letter]

PONE-D-24-27306R2

PLOS ONE

Dear Dr. Altare,

I'm pleased to inform you that your manuscript has been deemed suitable for publication in PLOS ONE. Congratulations! Your manuscript is now being handed over to our production team.

Kind regards,

on behalf of

Dr. Sk Md Mamunur Rahman Malik

Academic Editor

PLOS ONE